# Where Do We Stand in the Behavioral Pathogenesis of Inflammatory Bowel Disease? The Western Dietary Pattern and Microbiota—A Narrative Review

**DOI:** 10.3390/nu14122520

**Published:** 2022-06-17

**Authors:** Iwona Krela-Kaźmierczak, Oliwia Zakerska-Banaszak, Marzena Skrzypczak-Zielińska, Liliana Łykowska-Szuber, Aleksandra Szymczak-Tomczak, Agnieszka Zawada, Anna Maria Rychter, Alicja Ewa Ratajczak, Kinga Skoracka, Dorota Skrzypczak, Emilia Marcinkowska, Ryszard Słomski, Agnieszka Dobrowolska

**Affiliations:** 1Department of Gastroenterology, Dietetics and Internal Medicine, Poznan University of Medical Sciences, 60-355 Poznań, Poland; lszuber@wp.pl (L.Ł.-S.); aleksandra.szymczak@o2.pl (A.S.-T.); a.zawada@ump.edu.pl (A.Z.); a.m.rychter@gmail.com (A.M.R.); alicjaewaratajczak@gmail.com (A.E.R.); kingskoracka@gmail.com (K.S.); emilamar@ump.edu.pl (E.M.); agdob@ump.edu.pl (A.D.); 2Institute of Human Genetics, Polish Academy of Sciences, 60-479 Poznań, Poland; mskrzypczakzielinska@gmail.com (M.S.-Z.); slomski@up.poznan.pl (R.S.); 3Doctoral School, Poznan University of Medical Sciences, 61-701 Poznań, Poland

**Keywords:** IBD, Western diet, Mediterranean diet, microbiota, nutrients, risk factors of IBD

## Abstract

Despite the increasing knowledge with regard to IBD (inflammatory bowel disease), including ulcerative colitis (UC) and Crohn’s disease (CD), the etiology of these conditions is still not fully understood. Apart from immunological, environmental and nutritional factors, which have already been well documented, it is worthwhile to look at the possible impact of genetic factors, as well as the composition of the microbiota in patients suffering from IBD. New technologies in biochemistry allow to obtain information that can add to the current state of knowledge in IBD etiology.

## 1. Introduction

Inflammatory bowel disease (IBD) is a chronic inflammatory disorder of the gastrointestinal tract. It is characterized by relapses and remissions, thus requiring lifelong treatment. The etiology of the disease is complex. Despite significant advances in the molecular biology and knowledge regarding IBD, the pathogenesis of the disease remains unclear. Since the microbial community has been found to play a huge role in the human health and has been proved to be altered in IBD patients, it seems that it could be one of the crucial elements of IBD development. In addition to the already well demonstrated immunological, genetic and environmental risk factors of IBD, in this review, we focus, in particular, on the complex structure of intestinal microbiota, its role in the host organism and the Western dietary pattern that may affect microbial composition and the risk of IBD. Moreover, we highlight recent advances in this area that may contribute to the discovery and understanding of the complete picture of IBD pathogenesis, as well as providing new ideas for IBD therapy. 

## 2. Inflammatory Bowel Disease

Inflammatory bowel diseases (IBD), which include ulcerative colitis (UC) and Crohn’s disease (CD), are among disorders with still undetermined etiology. They comprise numerous immunological, genetic, microbiological, environmental and dietary factors [1]. In recent years, progress has been observed in the search for the genetic factors associated with IBD. In fact, it has been proved that one of the elements affecting the functions of the immune system are genetic mutations responsible for the expression of numerous proteins that have a regulatory effect. However, studies of monozygotic twins demonstrate low compliance rates in the incidence of these diseases, in CD (20–55%) and UC (6.3–17%), which in turn confirms the hypothesis that other coexisting factors must be involved. More than 200 genes have been discovered that predict the development of IBD. Most of them are genetic polymorphisms associated with the function of the mucosal barrier, responsible for the direct interaction with gastrointestinal antigens, including the microbiota and dietary components [2,3,4,5]. Furthermore, an epidemiological study has indicated that significantly more cases of IBD are observed in highly developed countries. It is also worth noting that first-generation immigrants who had moved to these countries were at a higher risk of IBD [6]. These observations confirm the undeniable influence of environmental factors on the development of the afore-mentioned diseases, and they impact the gastrointestinal microbiota together with dietary factors. Nowadays, the microbiota is seen as an integral part of keeping the human body healthy. A microbiome is a collection of microbes genes in the host organ [7]. In many investigations, it has been proved that disorders in the composition of the intestinal microbiota, defined as a complex of microorganisms inhabiting the intestines, directly affect the cells of the immune system, thus stimulating its activity and inducing the activity of inflammation in the mucosal membrane. In fact, some reports indicate that patients with IBD present a characteristic dysbiosis [8,9]. The role of the intestinal microbiota in the pathogenesis of IBD may also be confirmed by the effect of antibiotic therapy, probiotic intake and the beneficial effect of fecal microbiota transplantation on the induction of disease remission [10]. The most relevant findings related to IBD are shown in Figure 1 (based on Mulder et al.) [11]. 

Although diet seems to play a significant role in the pathogenesis of IBD, epidemiological studies regarding its impact on these disorders are challenging to perform due to a large number of confounding factors. However, based on the research conducted to date, it can be concluded that a specific nutritional profile has a beneficial effect on the course of IBD. An increased intake of unsaturated fatty acids, fruits, vegetables and breastfeeding undoubtedly shows a protective effect on the intestines. Recently, much attention has been paid to the Mediterranean diet, and it seems that its promotion contributes to reducing the risk of developing IBD [12,13].

## 3. Risk Factors of IBD

### 3.1. Genetics

The basic genetic knowledge of the IBD hereditary character stems from comparative studies that investigated the prevalence of the disorder in first-degree family members, relatives of the affected individuals, as well as from twin studies. In 2002, research results conducted on 1000 IBD patients over the period of 20 years were described [14]. As many as 14% of respondents reported at least one family member in the first line with a CD in their family history. This is consistent with the studies conducted by other researchers who estimate the group of CD patients in the family history at 10–20% [15]. Having a first-degree relative family member diagnosed with CD increases the risk of developing the disease 15–35 times, whereas, in the same circumstances, the risk for UC is increased only 6–9 times [16]. Observations of monozygotic twins with CD proved that the incidence rate is 20–58.3% (average 30.3%). In fraternal twins, it does not differ from the incidence observed in non-twin siblings and is equal to approximately 3.6%. In terms of UC, the correlation is much weaker and amounts to 6.3–18.2% (average 15.4%) [17,18,19].

The first study indicating the *locus* associated with CD development was published in 1996 by Hugot and co-authors. The location at the band 12 in the short arm (q) of chromosome 16 (16q12) was then defined as the IBD1 region [20]. In 2001, the same research team achieved another success by narrowing down the IBD1 area to a specific *NOD2* gene and three major mutations: rs2066844 (c.2104C > T, p.Arg702Trp in exon 4), rs2066845 (c.2722G > C, p.Gly908Arg in exon 8) and rs2066846 (c.3019_3020insC, p.Leu1007fs and in exon 11) associated with CD diagnosis within this gene [21,22] (Figure 2). In fact, these variants may increase the risk of developing the disease up to 40 times [23]. The *NOD2* gene (formerly known as CARD15) encodes an intracellular receptor that regulates the non-specific immune response, expressed in peripheral blood monocytes, macrophages and intestinal mucosa cells, particularly Paneth cells. The NOD2 protein, consisting of two N-terminal domains (CARD), a nucleotide-binding domain (NBD) and a C-terminal end containing ten leucine-rich repeats (LRR), is involved in bacterial pathogen recognition and inflammasome initiation via binding to muramyl dipeptide. To date, *NOD2* represents the best-known gene with regard to the involvement of genetic factors in CD [24]. The recessive inheritance of rare and low-frequency deleterious *NOD2* variants accounts for 7–10% of the CD cases [25]. Moreover, the relationship between *NOD2* mutations and the location of the disease in the ileum area has been proved. However, differences in the occurrence of the mutations in the region of this gene have been observed, e.g., the Japanese, Chinese, and Koreans are the populations free from *NOD2* variants, whereas the changes occur very rarely in the African Americans suffering from IBD [26,27].

According to data from the Online Mendelian Inheritance in Man (OMIM) database, to date 28 regions (IBD1-28) have been defined in the human genome containing specific genes and changes that are associated with an increase in IBD incidence (Table 1).

The list of DNA *loci* that may be related to the pathogenesis of IBD is much longer and includes nearly 300 candidate genes, mainly as a result of genome-wide association studies (GWAS) [28,29,30,31]. In addition to the DNA sequence variants, epigenetic research is also a new, promising research area in IBD. DNA modifications appear to be causally involved in the pathogenesis of IBD, as evidenced by studies of CD8 + T-cell in vitro cultures obtained from IBD patients [32], or genome-wide analysis of the DNA methylation profile [33]. Such epigenetic research opens up a broad field of research that generates exciting insights into IBD pathophysiology. The patterns of DNA methylation and histone modification as well as microRNA (miRNA) molecules post-transcriptional regulation may in the future serve not only as biomarkers of disease predisposition, disease activity, or disease course, but also as new targets in therapeutic interventions in IBD patients [34,35].

The genetic factors involved in IBD represent a comprehensive issue. In this paper, which focuses on dietary patterns and microbiota in IBD patients, it is worth emphasizing that the subject of recent research is genetic defects affecting the homeostasis of the intestinal barrier. In particular, the impaired expression of genes related to the cellular involvement, junction complexes, mucus production and secretion, pathogen detection, Paneth cell activity, reactive oxygen species production, xenobiotic responseand IgA secretion severely impairs the intestinal epithelial barrier integrity and protective functions in IBD patients [36]. For instance, studies on a genetically engineered mouse model have shown that the dysregulated expression of genes encoding commitment-related transcription factors, such as SRY-box transcription factor 9 (*Sox9*), hairy and enhancer of split 1 (*Hes1*), serine-threonine kinase 11 (*Stk11*), mouse atonal homolog 1 (*Math1*), caudal type homeobox 2 (*Cdx2*) and GATA-binding factor 6 (*Gata6*), definitively compromises intestinal epithelial cell differentiation and, as a consequence, intestinal barrier integrity with its secretive (particularly Paneth cells) and adhesion functions [37,38,39,40]. Moreover, in Crohn’s disease patients, the impairment of epithelial junctional complexes was found to significantly contribute to the development of chronic inflammatory conditions, which may be caused by an increased expression of claudin-2 (*CLDN2*) and an impaired expression and redistribution of claudin-5 (*CLDN5*), claudin-8 (*CLDN8*) and occludin (*OCLN*). Hence, an increase in intestinal permeability and bacterial translocation is observed [41]. Similar relationships have been described in the studies of ulcerative colitis patients in whom the deregulated expression of *OCLN*, *CLDN1*, junctional adhesion molecule 1 (*F11R*, also known as *JAM*), beta-catenin (*CTNNB1*) and epithelial cadherin (*CDH1*) led to the transepithelial migration of neutrophils [42]. Moreover, animal studies confirm these observations, e.g., that mice with a knockout of the *F11r* gene, despite having a normal structure of the intestinal epithelium, developed low-grade colonic inflammation [43].

O-glycans of the gastrointestinal tract constitute the first defense line against external stimuli. Defects in O-glycosylation may lead to damage in mucin expression classified into three categories: secretory gel-forming (MUC2, MUC5AC, MUC5B and MUC6), secretory non-gel-forming (MUC7) and membrane-bound (e.g., MUC1, MUC3, MUC4, MUC12, MUC13 and MUC17). Consequently, mucosal barrier destruction occurs, leading to the microbial activation of inflammasomes [44]. Particularly, *Helicobacter pylori* and *Clostridium difficile* infections are associated with disrupted mucin synthesis and mucus barrier, whereas *Helicobacter pylori*-infected patients present with a significant decrease in *MUC5AC* gene expression levels [45].

In conclusion, it should be noted that a new area in the study of genetic factors of IBD has recently been established. Haberman et al., in their observations, demonstrated a significant reduction in the expression of mitochondrial genes encoding the oxidative phosphorylation chain and nuclear genes in patients with UC [46]. Researchers have suggested and simultaneously confirmed in preliminary studies that mitochondriopathy is also a process associated with the pathogenesis of IBD [47].

### 3.2. Immunology

Research with regard to intestinal immunology has long played a fundamental role in understanding the pathogenesis of IBD and has primarily focused on investigating the mucosal response of the intestines. It has been suggested that disorders in innate immunity, as well as adaptive immunity, contribute to the development of IBD. 

The adaptive response is specific and continues for several days; it is also predominantly T-lymphocyte dependent. Most researchers investigating this type of response favor the theory that CD is a disease associated with increased Th1 lymphocyte activity, while UC entails the activity of Th2 lymphocytes [48,49]. Th1 cells produce large amounts of IFNγ, IL-2 and TNFα, and Th2 cells release IL-4, IL-5 and IL-13. Some studies have indicated that, in cells grown in vitro for both CD and UC, exact high amounts of IFNγ are observed [50]. Furthermore, research conducted in recent years has suggested that Il-13 has anti-inflammatory effects [51]. Wilson et al. observed that supernatants from biopsies grown in vitro for both UC and CD have lower IL-13 concentrations compared to IFNγ. In addition, researchers have found that the concentrations of these cytokines are comparable in both diseases [52]. Therefore, some researchers believe that both forms of Th-lymphocyte response coexist in patients with IBD. In terms of adaptive response, it is essential to take into consideration the Th17 cell. High concentrations of IL-17A, produced by these lymphocytes, were found in both the mucous membrane of patients with CD and UC in comparison to the control group. In fact, Th17 cells also constitute an important source of IL-21, a cytokine associated with the activity of IL-2, one of the primary inflammatory cytokines in the intestine. It has also been shown that there is an imbalance in IBD between Th17 lymphocytes and regulatory T (Treg) lymphocytes. Th17 cells activate the inflammatory process, and Treg cells inhibit autoimmunity in IBD. To maintain the proper balance of these cells, factors that stimulate cytokine signaling, such as intestinal microflora or bile acid metabolites, also play a role [53]. Much attention has also been paid to innate forms of the immune response, such as the integrity of the epithelial barrier, the reaction to the intestinal microbiota, and food antigens derived from its lumen. This form of response is our first line of defense against pathogens—thanks to it, unlike the adaptive response, the body reacts quickly to stimuli within minutes or hours. The first barrier to food and bacterial antigens coming from the intestinal lumen is its mucous layer. The second line is the intestinal epithelium, consisting of enterocytes and other specialized cells. In addition to mucus, these cells can produce a number of antimicrobial peptides. The defective expression has been proved in patients with CD [54]. The second line of defense consists of numerous cells, such as epithelial cells, macrophages, NK cells, monocytes and dendritic cells. Their activation occurs via Toll-like receptors (TLR) located on the surface of cells, and NOD-like, located in the cytoplasm. In patients with IBD, we observe the destruction of these receptors and effector cells [55]. A key cytokine that plays a role in the early response to microbial antigens is IL-23. Its polymorphisms are associated with both CD and UC, making it a typical cytokine in inflammation activity in IBD. IL-23 induces Th17 cells and cells of the innate immune system [56]. 

Many studies focus on mutations in the NOD-2 receptor range. These mutations are primarily associated with susceptibility to CD. Their consequence is an abnormal response of the immune cells of the intestine to bacterial lipopolysaccharides (LPS). There are various hypotheses as to the role of these mutations. It is believed that it may contribute to the lack of inhibition of TLR2 receptors, which results in the uncontrolled development of the inflammation and activation of Th-1 lymphocytes. Another theory is related to the reduced activation of NFkB, associated with a reduced reaction of the immune system cells to bacterial antigens [57]. 

An essential element of maintaining cellular homeostasis and an element of the host’s defense against foreign antigens is autophagy, which consists of the controlled decomposition by cells of chemical molecules, cell organelles and other cell fragments. The protein complex ATL16L1 (autophagy-related 16-like 1) is responsible for the proper phenomenon of autophagy. It has been shown that, in patients with IBD, there is a mutation in the *ATL16L1* gene, which results in the excessive elimination of intestinal epithelial cells and, consequently, increased permeability [58]. 

Immunological factors associated with Crohn’s disease and ulcerative colitis are presented in Table 2. 

### 3.3. Environment

Several epidemiological and laboratory data demonstrate that the etiology of IBD is the result of an interplay of genetic, immunological and environmental factors. Environmental factors associated with IBD include smoking; post-appendectomy status; the use of oral contraceptives, antibiotics and non-steroidal anti-inflammatory drugs (NSAIDs), which may affect the composition of the gut microbiome; diet; breastfeeding; and past infections, vaccinations, and environmental pollution [59,60,61]. Unfortunately, the results of many studies on the role of environmental factors in the pathogenesis of IBD often remain inconsistent. 

One of the best-studied factors is smoking. In his meta-analysis, Calkins showed that active smokers were less likely to develop ulcerative colitis compared to those who did not smoke or had stopped smoking. The study also showed that active and former smokers had a significantly increased risk of Crohn’s disease [62]. In addition, in patients with CD, smoking may be associated with an aggressive disease course, more surgical interventions and the need for intensified treatment [63,64]. It has also been shown that women with CD may be particularly adversely affected by smoking [65]. 

The exact mechanisms of the effects of tobacco smoke on the development of IBD are not well understood. One theory is that nicotine, via nicotinic acetylcholine receptors (nAChRs) present on T cells, may affect their function and cause the modulation of cellular immunity and the alteration of the cytokine profile [66]. Other hypotheses include altering the composition of the gut microbiome and generating oxidative stress due to free radicals [67].

Another important risk factor is having an appendectomy after the age of 20. It has been shown that the so-called early appendectomy is associated with a reduced risk of UC [68]. Data on the association of appendectomy with the development of CD remain conflicting. Frisch et al. and Koutroubakis et al. showed that appendectomy is a predisposing factor for CD [69,70], while Radford-Smith et al. report its protective role [71]. However, the mechanism of how appendicitis affects the risk of developing IBD is not yet fully understood. It has been postulated that bacteria in the appendix may play a role in the immune response to the intestinal microflora. 

Khalili et al. found an association between oral contraceptive use and IBD development, particularly CD [72]. It has been suggested that the estrogens in these drugs may enhance the humoral response and stimulate macrophages; progesterone, on the other hand, has strong immunosuppressive effects [73].

The interaction between the gut microbiome and the intestinal barrier has been shown to play an important role in the pathogenesis of IBD. Thus, the use of drugs, such as antibiotics and non-steroidal anti-inflammatory drugs (NSAIDs), that modify the composition of the gut microbiota may contribute to an increased risk of disease [74]. Aniwan et al., in a population-based case–control study, showed a strong association between antibiotic use and the risk of CD as well as UC. This risk was increased for all forms of IBD according to age [75].

Antibiotic use during childhood may be particularly important. It has been postulated that it may lead to the impaired development of gut bacteria tolerance, which is a direct risk factor for IBD [76]. Ananthakrishnan et al. suggest that the frequent use of NSAIDs, but not aspirin, seemed to be associated with an increased absolute incidence of CD and UC [77]. NSAIDs may cause direct damage to the gastrointestinal mucosa and, through cyclooxygenase inhibition, affect prostaglandin production and thus exert important immunoregulatory functions. 

The developmental hygiene hypothesis is also worth mentioning, which assumes that the increase in immunological disorders can be attributed to improved hygienic and sanitary conditions of the population and consequently less exposure to intestinal pathogens in childhood; it leads to an abnormal response to antigens later in life [78]. 

Water and air pollution associated with urbanization and industrial development may also contribute to the development of IBD. These include particulate matter, nitrogen dioxide, sulfur dioxide and pollutants from fossil fuel combustion, which can migrate across the gastrointestinal barrier and cause DNA damage and the disruption of immune mechanisms, which can result in the development of IBD [79]. 

Stress is also one of the important factors influencing the development of inflammatory bowel diseases [80,81]. It is well known that stress factors can affect immune functions via neuropeptides released from nerve cells, which in turn can lead to damage to the intestinal mucosa and thus impair its function as a protective barrier [82,83]. Bitton et al. suggest that individuals presenting lower stress levels show a reduced risk of Crohn’s disease [84]. Camara et al. emphasize above that the association between perceived stress and the exacerbation of Crohn’s was entirely related to mood components, specifically anxiety and depression [85]. 

It is worth mentioning that breastfeeding is a recognized protective factor for IBD. Human milk is believed to contain many substances that can affect growth and development and gastrointestinal function. In addition, the composition of the colonic bacterial flora has been shown to differ between breastfed and bottle-fed infants [86]. 

Understanding the influence of specific environmental factors on the development and course of IBD requires further detailed studies with different populations. Environmental factors influencing the risk of inflammatory bowel disease are presented in Table 3. 

### 3.4. Gut Microbiota

Over one hundred trillion various microorganisms, including bacteria, archaea, fungi, viruses and protozoa, colonize the human gastrointestinal tract (GI), constituting the gut microbiota [87]. This complex ecosystem plays a fundamental role in the host organism. Gut-associated lymphoid tissue (GALT), as part of the host immune system, is involved in the intestinal immune and metabolic responses via fermentation products, such as short-chain fatty acid (SCFA) and acetate, also in the regulation of host mucosal homeostasis, hormones release and vitamin synthesis. Thus, its impact on human health is crucial [88,89]. The power of this ecosystem also supports the fact that the collective genome of gut microbial flora is estimated to contain 100 times more genes than the human genome [90].

The gut microbiota also influences the immune system, which has a direct effect on the maintenance of the intestinal immune barrier and the development of inflammatory bowel disease. Moreover, the impact of both commensal bacteria and pathological biota on the immune system has been emphasized. Commensal microorganisms stimulate the maturation and differentiation of Treg lymphocytes, and affect the production of IL-10, which plays a key role in maintaining Th1/Th2 homeostasis; thus, they affect the development of the immunotolerance and maintenance of cytokine balance.

CD4 + T lymphocytes of the intestinal wall plaque secrete IL-17 and IL-22, which have a regulatory function in the development of the inflammatory process in the intestine, whereas IL-17 induces the expression of the chemokines CXC and CC [91,92]. In turn, Peyer cells contribute to the production of B cells and plasma cells, which produce IgA, protective of the intestinal barrier [93,94]. 

In germ-free mice, the mRNA expression level of the bactericidal protein Ang4, secreted by Penatha cells, is significantly lower compared to conventional mice, which also confirms the involvement of biota in the intestinal mucosal immune barrier function [95]. 

Additionally, the microbiota affects the regulation of the CARD9 gene responsible for inflammation in IBD [96].

In patients with moderate colitis, IL-22 production was elevated following the exposure to altered gut microbiota [97].

In IBD, the amount of SCFAs produced by the intestinal biota is significantly reduced [98]. In fact, SCFAs affect the immune system by binding GPR43 and provides the homeostasis of colonic Treg cells [99,100].

Research conducted in recent years has shown that disturbances in the composition of the gut microbial community (dysbiosis) and in its metabolic balance as well as, finally, the interaction of microbiota with the host are associated with IBD pathogenesis [8,101,102,103]. Although it is still not fully known whether these changes cause the disease or, rather, are a consequence of the intestinal inflammation. However, there is evidence that some microbial metabolites may directly affect the inflammation process via the expression of genes, such as anti- and pro-inflammatory cytokines, and can influence T-helper cell (Th) 1, 2 and 17 and regulatory T-cell (Treg) release [104]. In particular, commensal bacteria from *Clostridium* clusters can induce the differentiation of colonic Treg cells by producing SCFA [105]. Studies performed on germ-free mice revealed a reduced development of Th17 cells and a decreased number of lymphocytes and immunoglobulin (Ig) A in the intestinal mucosa of these animals compared to controls [106,107,108,109]. 

Numerous animal studies confirm the contribution of the intestinal microbiome in IBD pathogenesis. Researchers observed that a germ-free environment prevents UC in genetically susceptible mice [110]. Nevertheless, the transfer of dysbiotic bacteria to the healthy mice induces and develops inflammation [111]. Furthermore, the colonization of mice with gut microbiota obtained from IBD patients alters the immune response and exacerbates UC in mice [112]. Additionally, the transfer of naive CD4^+^ lymphocytes from the healthy mice to those lacking T and B lymphocytes may induce UC, wherein the susceptibility depends on the gut microbiota composition [113]. 

Multiple observations of human subjects also confirm the contribution of the microbial community in the mechanism of IBD induction. It has been observed that IBD activity is most apparent in the parts of the intestine where bacterial populations are high and at a standstill (colon and rectum). In CD patients, disease remission occurs in the segment of the intestine excluded from the fecal stream. Similarly, the post-operative re-exposure to the fecal flow in the intestine correlates with CD relapse [114]. 

Although the IBD pathogenesis has not yet been fully understood, new insights continue to emerge through research. The disease is a complex interplay of the environment, immune system and microbiota in a genetically susceptible host [115]. Table 4 shows the most important changes in the gut microbiota in inflammatory bowel disease. 

## 4. Characteristics of the Gut Microbial Community in IBD

### 4.1. Microbiota Composition

The metagenomic sequencing of fecal samples from healthy European subjects revealed that bacteria are the most numerous microorganisms, 90% to 99% of which belong to *Firmicutes* and *Bacteroidetes* phyla. The remaining ones include *Proteobacteria*, *Actinobacteria* and *Verrucomicrobia* [116]. Following the rapid advance of genomics, numerous global studies have described disturbances in the gut microbiota in IBD patients as compared to the healthy controls. However, they often produce inconsistent results, mainly due to differences in the methodology and the sample type (stool and mucosal biopsy) used. In view of the Genome Standards Consortium guidelines regarding reliable comparative studies on the microbiota composition, research should be based on the same methodology, including identical amplified regions of marker genes [117]. There are interesting common observations from the entire pool of available world studies, which confirm microbial pathogenesis in IBD [115].

Generally, the intestinal microbiota of IBD patients is less stable and demonstrates a significant reduction in microbial diversity. Additionally, dysbiosis is observed as a decreased abundance in commensal taxa and an increased abundance in pro-inflammatory and pathogenic microbes in bacterial, viral and fungal communities, as characterized in the integrative Human Microbiome Project (iHMP) [118,119]. 

Among the bacterial shifts at the phylum level, a decrease in *Firmicutes* abundance is most frequently reported in both CD and UC patients [120,121]. This is particularly relevant, since *Firmicutes* include beneficial, anti-inflammatory probiotic bacteria producing SCFAs, essential for the correct growth and differentiation of the epithelial cells. Thus, primary SCFAs producers, such as *Faecalibacterium prausnitzii, Clostridium XIVa, IV*, *Lactobacillus* and *Roseburia faecis*, are reported at reduced levels in CD and UC patients [122,123]. The depletion of “good bacteria” leads to impairment in mucosal permeability. *Faecalibacterium prausnitzii*, considered a biomarker of disease activity, recurrence and response to the treatment in CD and UC, has a particularly high prognostic significance [124]. In fact, reduced levels of *F. prausnitzii* in stools were predictive of CD relapse in patients in remission. Furthermore, these bacteria are associated with the secretion of metabolites that block the nuclear factor kappa-light-chain-enhancer of activated B cell (NF-kB) activation and IL-8 production by the intestinal epithelial cells [125]. *F. prausnitzii* is also protective in dinitrobenzene sulfonic-acid-induced colitis in mice [126]. Additionally, a unique protein produced by *F. prausnitzii*, referred to as microbial anti-inflammatory molecule (MAM), has been identified. This protein exhibits anti-inflammatory effects and inhibits the NF-kB pathway in epithelial cell lines [127].

*Firmicutes* levels are less frequently reported to be increased in IBD patients compared to the healthy controls, although it was only observed in a study performed on Polish UC patients [128,129]. An enrichment of *Ruminococcus gnavus* from the *Firmicutes* phylum and other mucolytic bacteria is found in IBD cases and leads to mucus deprivation and the enhancement of pathogenic bacteria invasion [118,130]. Most research conducted to date have also demonstrated a decrease in the *Bacteroidetes* phylum in UC and CD cases [121,129,131]. In contrast, an opposite outcome was obtained by Feuntes et al. in UC microbial samples [132]. 

The majority of the global studies have shown that IBD individuals present with an overgrowth of aerotolerant phyla, *Proteobacteria* and *Actinobacteria* [129,133]. This phenomenon may be accounted for by inflammation as an oxidative state that may increase the number of bacteria that adhere to the colonic mucosa and have a higher oxygen tolerance [134]. This group of microbes includes, for instance, *Escherichia coli* and sulphate-reducing *Desulfovibrio*, which have been shown to demonstrate a pro-inflammatory effect on the mucosa, and their increased abundance correlates with IBD, which was also observed in a study on Polish UC individuals [129,135]. 

The gut microbial flora of IBD cases is characterized by reduced *Verrucomicrobia* in comparison to the healthy populations. Furthermore, a commensal bacterium from this phylum, *Akkermansia muciniphila,* which could reduce the substance triggering colitis by derived extracellular vesicles, is also deficient in IBD. Its deficiency may result from reduced levels of mucin in patients with UC, which constitute an energy source for these bacteria [136]. In addition, recent studies have indicated *Akkermansia muciniphila* as a potential biomarker of IBD [124]. These bacteria were shown to improve the gut barrier via its outer membrane protein Amuc_1100, which interacts with Toll-like receptor 2 [137].

Interestingly, the CD microbial profile is more altered and unstable than that of UC [138]. In fact, there are eight bacterial groups that change specifically in CD in comparison to the healthy controls: *Anaerostipes*, *Methanobrevibacter*, *Faecalibacterium*, *Peptostreptococcaceae*, *Collinsella* and *Christensenellaceae* are decreased, and *Escherichia* with *Fusobacterium* are increased [138,139]. 

Moreover, intestinal microbiota composition significantly differs in CD depending on the location and activity of the disease. The colonic CD is characterized by a higher abundance of *Faecalibacterium*, *Bifidobacterium* and *Collinsella*, whereas ileal CD is associated with a decrease in *Faecalibacterium* and *Collinsella*, as well as with a higher abundance of *Enterobacteriaceae* family [140]. However, in patients with inactive CD, the enrichment of beneficial microbes, such as *Akkermansia*, *Barnesiella*, *Oscillibacter*, *Roseburia* and *Odoribacter*, was reported when compared to the active disease [141].

The conducted research demonstrated a correlation between oral and gut microbiota composition. Oral inflammation, such as periodontitis, exacerbates gut inflammation by supplying the gut with colitogenic pathobionts and pathogenic T cells [142]. Furthermore, viruses infecting prokaryotes and eukaryotes constitute a part of the gut microbial community. Additionally, changes in the virus composition are also observed in IBD, and correlate with bacterial alterations [143]. In fact, the increased abundance of the *Caudovirales* phage in the intestine correlates with a decreased bacterial diversity and mucosal inflammation [144]. According to the studies, the fungal microbiota in the human intestine, mainly composed of the *Ascomycota* and *Basidiomycota* phyla, plays a role in IBD pathogenesis [145]. Moreover, protective *Saccharomyces cerevisiae* is decreased, whereas *Candida albicans*, *Candida tropicalis*, *Clavispora lusitaniae*, *Cyberlindnera jadinii* and *Kluyveromyces marxianus* are increased in IBD subjects compared to the healthy controls [146]. In addition, the level of *C. albicans* has been associated with the remission and active phase of the disease.

### 4.2. Genetic Identification of Microbiota

Tremendous advances in DNA sequencing technologies and bioinformatics in the last two decades have transformed classical microbiology and allowed for studying the composition of entire microbial communities from various environments. 

The first report with regard to microbes was made in 1676 by Leeuwenhoek. A Dutch naturalist, referred to as the father of microbiology, observed living organisms in the plaque under a self-constructed microscope [147]. Since then, the most renowned scientists attempted to isolate these minute organisms until 1888, when Robert Koch began to culture and isolate them using solid nutrients [148]. The microscope constituted a fundamental tool to observe the interactions between microorganisms, and staining methods were developed to improve their resolution. The observation that microorganisms require a specific environment in order to grow constitutes a milestone in the development of microbiology, and provided the foundation for microbial ecology [149] (Figure 3). 

Another breakthrough in microbial studies occurred in 1977 with the proposition of ribosomal RNA (16S rRNA) genes as molecular markers for the classification of organisms [150]. The properties of this gene—having relatively short sequences, being highly conserved within a species and the differences between them—established it as an ideal taxonomic marker. Nine hypervariable regions (V1–V9) in the 16S rRNA gene were located, all of which could be involved in the taxonomic classification. Simultaneously, Fred Sanger developed an automated sequencing method [151]. The combination of both discoveries revolutionized the study of microorganisms. However, research showed that various 16S rRNA sequences do not belong to any known cultured species, indicating a massive number of non-isolated organisms. Therefore, the analyses of 16S rRNA genes performed in samples collected directly from the environment showed that culture-based methods found less than 1% of bacterial and archaeal species in the sample [152]. These conclusions contributed to a greater interest in microbial detection studies and the development of new molecular methods to identify microbiota [135,136,137,138].

Nowadays, NGS technology constitutes a basic tool used in the genetic identification and analysis of microbial communities. Two approaches for determining microbial diversity are currently available: metaprofiling and shotgun metagenomic sequencing [139,140,141,142]. Both strategies exhibit enormous potential and are effective not only in scientific research, but also clinical practice. However, in studying microbiota, it is crucial to follow the principles of good practice developed by the Genome Standards Consortium, particularly in terms of comparative metagenomics [117]. 

## 5. Nutrients of the Western Diet in the Course and Development of IBD

### 5.1. High Intake of Saturated Fatty Acid and Fast Food

The Western diet can be characterized as a high-fat diet (HFD), particularly rich in saturated fatty acids (SFA) and trans fatty acids. SFA can be found mainly in animal-derived products, such as butter, red meat and full-fat dairy, as well as some plant oils, including coconut oil or palm oil [153]. It seems that dietary fats play a role in the pathogenesis of IBD [154]. 

Studies on mice models indicate that HFD impacts the microbiota composition, thus inducing low-grade inflammation and increasing intestinal permeability and oxidative stress [154]. Furthermore, HFD exacerbates the severity of dextran sodium-sulfate-colitis through intestinal barrier disruption, the upregulation of pro-inflammatory cytokines production and an increase in oxidative stress in the colon tissue [155,156,157,158,159]. 

Enos et al. evaluated the impact of a diet composed of the same (40%) total fat content, but varying in the amount of saturated fat. Interestingly, the greatest insulin resistance, adiposity and macrophage infiltration were observed in the diet containing 12% saturated fats, while 24% and 6% saturated fat diets were found to demonstrate significantly lower rates [160].

In IL-10-knockout mice genetically predisposed to IBD, a diet rich in saturated fatty acids and low in polyunsaturated fatty acids derived from vegetable oils promoted dysbiosis with the proliferation of the sulfite-reducing pathobiont *Bilophila wadsworthia*. In fact, the consumption of saturated fatty acids was associated with the inflammatory response mediated by Th1 cells and colitis [161]. In another study, the authors compared secretory immunoglobulin A (sIgA) levels in mice fed with a normal-fat diet, as well as in mice fed with a HFD. As a result, a significant decrease in the level of sIgA coating the gut microbiota was observed in HFD-fed mice. Furthermore, data from this study provided insight into changes in the microbiota composition of HDF-fed mice. The researchers indicated that a decreased level of sIgA may result from a reduced microbial diversity and the concentration of SCFAs in HFD-fed mice [162]. 

A prospective study including 412 patients suffering from UC in clinical remission demonstrated an association between a high dietary intake of myristic acid and disease exacerbation. Myristic acid is a saturated fatty acid found in coconut and palm oils and dairy products [163]. 

Furthermore, Niewiadomski et al. conducted a study where the impact of the environmental exposure on IBD development was analyzed. They observed that frequent fast food intake, defined as more than once a week, was strongly associated with a 43% higher risk of UC and 27% higher risk of CD [164]. In addition, a Swedish study confirmed the link between consuming fast foods and a higher risk of developing UC and CD [165].

The daily consumption of fast food was associated with young age at the time of diagnosis in both CD and UC, as well as with an increased risk for surgery and severe disease course in UC patients [59].

### 5.2. Low Intake of Unsaturated Fatty Acid

Unsaturated fatty acids are mainly found in plants (e.g., vegetable oils, nuts, seeds or avocado) and animal-based dietary products, such as fish oils, and are divided into MUFA and PUFA. Omega-3 (n-3) PUFAs exhibit inflammatory properties, whereas n-6 PUFAs are described as pro-inflammatory compounds; thus, the ratio of n-6/n-3 PUFA intake appears to be important [166]. 

An interesting study was performed by Park et al. on mice with dextran sodium-sulfate-induced colitis. The authors investigated the effects of olive oil, coconut oil, corn oil and cottonseed oil intake on colitis development. They observed that the treatment with cottonseed oil had the most protective effect against intestinal inflammation by reducing the expression of inflammatory cytokines, oxidative stress markers and the development of intestinal fibrosis compared to olive oil, coconut oil and corn oil [167]. 

Furthermore, animal studies indicate n-3 PUFAs affect the development of CD [166]. 

Shoda et al. analyzed the correlation between CD incidence and dietary intake. They observed that animal protein and n-6 PUFA with less n-3 PUFA may contribute to the development of CD [168]. Additionally, the prospective Nurses’ Health Study cohort showed that a higher long-term intake of n-3 PUFA was associated with a lower risk of UC, whereas a higher intake of trans-unsaturated fatty acids was linked to an increased risk of developing UC. However, in this study, cumulative energy-adjusted total fat intake did not affect the development of UC or CD [169]. Moreover, a case–control study in a pediatric population indicated that n-3 PUFA and a higher ratio of n-3/n-6 were associated with a reduced risk of CD [170]. Tjonneland et al., in a prospective cohort study, indicated that a higher intake of n-6 linoleic acid increased the risk of developing UC [171]. Furthermore, the authors observed a 77% risk reduction in developing UC in patients who consumed the highest amount of DHA, omega-3. They did not observe a decreased risk of UC in patients consuming high amounts of ALA [171]. Similarly, John et al. observed that total dietary n-3 PUFAs, EPA and DHA, particularly DHA, were associated with UC protection in adults aged 45 [172]. 

Merton et al., in their meta-analysis, concluded that n-3 PUFAs reduced the level of pro-inflammatory cytokines, decreased CD activity and elevated the quality of a patient’s life. However, the authors simultaneously stated that there were numerous limitations in the 15 included studies [173]. Subsequently, Mozaffari et al. published a meta-analysis of observational studies in which they showed a negative association between fish consumption and the incidence of CD, as well as a significant inverse association between dietary long-chain n-3 PUFAs and the risk of UC [174].

In contrast, a meta-analysis of RCTs investigating the long-term effects of n-3, n-6 and total PUFA on IBD and inflammatory markers suggested that the supplementation with PUFA had little or no effect on the prevention or treatment of IBD [175]. Moreover, the meta-analysis by Wand et al. suggested a lack of association between fat intake and UC risk [176]. 

Another study showed a positive correlation between n-3 PUFA and CD risk, which, as the authors explain, may be due to a higher intake of the total fat, which potentially triggered IBD [177]. 

Bearing in mind the abovementioned data, studies on the impact of unsaturated fatty acids on IBD development provided mixed results. Thus, there is a need for further studies performed on larger groups of patients with the use of appropriate methodology. 

### 5.3. High Intake of Carbohydrates, Sweets and Sweet Drinks

Preda et al. suggested sweets and ice cream, as well as processed food, fried food, salt margarine, high-fat meal, mayonnaise and snacks, increased the risk of IBD in Romanian and Belgian IBD patients [178]. Possibly, sucrose increased the likelihood of IBD through its impact on the gut microbiota [179]. According to a multicenter case–control study, a higher intake of sweeteners and sugar increased CD risk. Moreover, the intake of vitamin C, mainly found in fruits and vegetables, reduced the risk of UC. It is vital to note that no associations were found between the total carbohydrate intake and the risk of CD or UC [177]. Similar results were observed in a meta-analysis, i.e., a factor associated with an increased risk of CD was sucrose intake, but not carbohydrates [180]. Furthermore, an EPIC (European Prospective Investigation into Cancer) study also reported that a high intake of sugar and soft drinks and a low intake of vegetables was associated with the risk of UC development [181]. Moreover, the meta-analysis of prospective cohort studies demonstrated that sugar intake was associated with an increased risk of IBD, UC and CD, although such observation was not found in terms of total carbohydrate intake and soft drinks [182]. 

A low-FODMAP (fermentable oligosaccharides, disaccharides, monosaccharides and polyols) diet may be useful, particularly in patients suffering from irritable bowel syndrome. Nevertheless, a study revealed that a low-FODMAP diet may also be used among patients with IBD. In fact, enteral nutrition with low FODMAPs reduced diarrhea compared with a high-FODMAP diet [183]. Conversely, Cox et al. reported that a relatively high dose of fructans, but not galactooligosaccharides and sorbitol, exacerbated gastrointestinal symptoms among patients with IBD [184]. 

### 5.4. Low Intake of Fibre, Whole Grain, Vegetables and Fruits

An American study showed that patients with IBD more frequently ingest less than 16.7 g of fiber per day than healthy adults. However, it is crucial to note that IBD patients more frequently consume vegetables, but not fruits, per day than the healthy individuals [185]. Fiber intake has been associated with a reduced risk of exacerbating CD, although not UC [186]. In contrast, Davies and Rhodes assessed relapse rates in UC patients treated with sulfasalazine with and without nutritional intervention (increased fiber intake). The authors reported that relapse rates in terms of standard and high fiber diets were 20% and 75%, respectively [187].

Moreover, a systematic review showed a negative association between fiber and fruit intake and the risk of CD development. Additionally, the intake of vegetables decreased the risk of UC [188]. However, a prospective cohort study did not demonstrate an association between fiber, vegetables, fruit and relapse in UC patients [189]. The results of a European prospective multi-center cohort study did not support the hypothesis that fiber intake was associated with the etiology of UC. 

According to a study based on a self-administered questionnaire completed by IBD patients, fruit, vegetables and bread with gluten resulted in the exacerbation of symptoms [190]. The meta-analysis showed that a higher intake of vegetables and fruits was negatively associated with UC risk. Additionally, fruit intake reduced the risk of CD. Vegetable intake was inversely associated with the risk of CD, although only in the case of studies performed in Europe [191]. Sandefur et al. described the case of an adult male with newly diagnosed CD who did not enter clinical remission despite standard treatment. Clinical remission was achieved when the patient followed the diet based only on grain, legumes, vegetables and fruits without medications [192]. 

## 6. Effects of the Western Diet on the Gut Microbiota Composition in IBD Patients

The gut microbiome has a very complex function in the human body. By influencing the digestion of food substrates and the production of vitamins and nutrients, it affects the metabolism. Furthermore, by means of modulating the immune system, influencing the production of SCFA, hydrogen peroxide and bacteriocin, it inhibits the expansion of pathological organisms, as it promotes the colonization of beneficial biota in the intestine [193,194]. 

Western diets consist mainly of high-calorie, highly processed foods, and the main components are saturated fats, high-glycemic-index carbohydrates and animal proteins. Through changes in the gut microbiota, this diet shows direct pro-inflammatory effects within the intestine and immune system, which may be related to the development of IBD [100,195]. The adverse effects of a high-protein diet were revealed by the fact that unabsorbed proteins reach the colon, affecting the microbiota by means of reducing the number of butyrate-producing *Roseburia/E. Rectale* [196]. In addition, catabolism results in excessive amounts of gases, such as hydrogen sulfide and ammonium, which may affect the intestinal barrier and damage colonocytes [197]. 

Similarly, the high carbohydrate intake in Western diets, e.g., high intake of glucose and fructose, results in dysbiosis, which affects intestinal permeability and increases inflammation. The high- fat intake in Western diets also has an impact on the microbiota in IBD patients. The study by Jee-Yon Lee found that antibiotic use in individuals on a high-fat diet was associated with the highest risk of IBD in mice [198]. Additionally, it was associated with impaired mitochondrial bioenergetics and increased intracellular lactate in colonocytes. In contrast, PPAR-γ signaling activated by the microbiota is a homeostatic pathway and prevents the expansion of potentially pathogenic *Escherichia and Salmonella* by reducing the bioavailability of the respiratory electron acceptor *Enterobacteriaceae* in the colonic lumen [199]. 

In addition, the long-term consumption of high fat and sugar has been shown to increase dysbiosis by increasing the number of *Bacteroides* spp. and Ruminococcus [200]. A comparison of the Western diet with the Eastern diet based on plant carbohydrates revealed that the Eastern microbiota showed a higher prevalence of Prevotella spp. instead of *Bacteriodes* spp. [201]. Carbohydrate sources of protein and fat were associated with higher numbers of *Bacteriodes* spp., and the prevalence of simple carbohydrates and dietary fiber were mainly associated with the increase in *Prevotella* spp. [202].

## 7. Is the Mediterranean Diet an Alternative to the Western Diet in IBD Patients?

The Mediterranean diet (MD) is a dietary pattern characterized by a high intake of fresh fruits and vegetables, legumes, whole grain cereals, nuts, moderate consumption of dairy, meat and fish, and the abundant use of olive oil with the moderate consumption of dry, red wine [203]. Its beneficial effect with regard to various conditions, such as obesity, metabolic syndrome, cardiovascular disease or diabetes, was proved both in observational and clinical studies [204,205,206,207,208]. 

However, its high content of anti-inflammatory and antioxidative bioactive compounds, e.g., mono- and polyunsaturated fats, polyphenols or vitamin E, affects other diseases in which a high intake of anti-inflammatory nutrients could be beneficial, and is also widely investigated [209,210,211,212]. 

The main source of bioactive nutrients in MD seems to be olive oil (80% of monounsaturated fatty acids and the presence of phenolic fractions), in particular extra virgin olive oil, which could provide a therapeutic option in IBD individuals due to the high phenol content. However, further studies are necessary in order to develop specific strategies and recommendations [213]. 

Contrary to the Western-style diet, MD helps to maintain the intestinal barrier integrity and healthy intestinal microbiota, preventing systemic endotoxemia and chronic inflammation [214,215]. 

As part of a healthy lifestyle, physical activity is an essential part of everyday activities. According to the study by Bibiloni et al., MD was associated with more leisure-time physical activity, whereas WD was associated with a slower gait speed and poorer body strength, agility and aerobic endurance [216]. In fact, similar results were observed in other studies, involving different age groups [217,218,219]. 

Although MD may be an inappropriate dietary approach in the disease flare-up, due to the high content of dietary fiber, taking the current data into account, it should be recommended during remission (possibly a more tailored option). Moreover, IBD patients may experience nutritional deficiencies, mainly in terms of micronutrients; thus, they should follow a nutrient-rich diet, such as MD (contrary to the WD) [220,221]. Furthermore, several studies have shown that individuals with IBD have an increased cardiovascular risk compared to the general population, promoting the MD approach in the afore-mentioned group [222]. Nevertheless, data regarding MD consumption in individuals with IBD are currently limited. 

In the study of Khalili et al., a greater adherence to MD was associated with a significantly lower risk of later-onset CD (age, gender, education level, body mass index and smoking did not modify these associations), but not UC [12]. On the other hand, WD was associated with a greater likelihood of CD development and a carnivorous pattern with UC development [223]. Additionally, adherence to MD was associated with an improvement in quality of life among individuals suffering from CD (IBDQ score) [224]. Moreover, a significantly higher MD score was verified in patients with inactive CD, rather than in individuals with the active disease, and the MD score was negatively associated with CRP concentrations. Individuals with inactive CD had significantly higher protein and vitamin C intake levels than patients with the active disease. In another study by Chicco et al., the 6-month consumption of MD significantly reduced malnutrition-related parameters, liver steatosis and quality of life among individuals with CD and UC [225]. 

In contrast, the EPIC study did not indicate an association between MD and the risk of CD [181]. Interestingly, in the same cohort, dietary oleic acid (e.g., found in olive oil) was inversely associated with UC development (dose-responsive, lower odds ratios with oleic acid intake equal to 21.13–40.93 g/day) [226]. In the study of Morvaridi et al., the consumption of extra virgin olive oil (20 days, 50 mL) significantly decreased inflammatory markers (CRP and erythrocyte sedimentation rate) and improved gastrointestinal symptoms (including bloating, fecal urgency and incomplete defecation) in patients with UC [227]. However, it is worth bearing in mind that the content of extra olive oil’s bioactive compounds can differ between olive oils due to the environmental conditions, extraction procedures, harvesting handlings and storage. Therefore, good-quality, non-refined olive oils stored in cold, dark places (as well as in dark bottles) should be recommended [228].

Another vital issue in the dietary recommendations is the ease and willingness to adhere to a specific dietary approach. In the study of Vrdolijak et al., adherence to MD was very low, particularly regarding olive oil, fresh fruits and vegetables, although IBD individuals were willing to extend their nutritional knowledge if proper educational programs were organized [229]. According to the study by Fiorindi et al., adherence to MD was also affected by disease activity [230]. Among CD patients, a higher MD adherence was observed in individuals with the inactive disease and in patients who had undergone two or more surgeries (no significant differences in both aspects were observed in individuals with UC). Moreover, MD may modify intestinal inflammation in the IBD group, as shown in the study of Godny et al., where MD adherence was associated with decreased calprotectin levels in patients with UC following pouch surgery [231]. Additionally, the group with a regular pouch higher adherence to MD tended to be inversely associated with the onset of pouchitis [231]. A summary of the nutritional factors affecting the risk and/or the exacerbation of inflammatory bowel diseases are shown in Table 5. 

## 8. Dietary Recommendations for IBD Patients That Are Beneficial for the Gut Microbiota

Dietary fiber represents a major component of plant-based diets, which is not very common in the Western diet. It may modulate intestinal inflammation and play a protective role against IBD by means of shortening the intestinal transit, synthesizing SCFAs (acetate, butyrate and propionate) and having beneficial effects on intestinal microbiota composition and the intestinal barrier [232]. However, since it may exacerbate gastrointestinal symptoms (bloating, constipation and diarrhea in patients with IBD even during remission), it should be supplemented with fiber-degrading bacteria in patients suffering from IBD, and used in amounts tailored to the individual requirements [233]. In remission, IBD patients should follow a low insoluble fiber diet with most of the soluble fiber fraction and with a high content of vitamins and minerals. 

Individuals with IBD often experience vitamin deficiency, particularly fat-soluble vitamins A and D, and water-soluble vitamins B12 and folic acid. In fact, studies have shown that supplementation with vitamin A, a major antioxidant, reduces intestinal inflammation by protecting against free radicals [188,234]. 

Nevertheless, taking into account the beneficial effects of MD on the inflammatory status, oxidative stress, dietary intake, IBD-associated diseases and lifestyle in general (including quality of life and physical activity), MD should constitute an alternative to the WD in IBD despite insufficient data. It may be suggested that, even if MD could not be fully introduced (e.g., in non-Mediterranean countries or due to the gastrointestinal symptoms), efforts to fully adhere to MD should be sufficient enough in the prevention and treatment of IBD. 

## 9. Summary and Conclusions

Several concluding points may be outlined on the basis of the presented data. 

The current theory regarding the pathogenesis of IBD states that enteritis results from an abnormal immune response in genetically predisposed individuals with coexisting disorders in the composition of the intestinal microbiota [193].The role of genetic predisposition in IBD is unquestionable, although it has not yet been fully defined. The current knowledge is incomplete with regard to the interdependencies and interactions of IBD factors, including genetics, their interactions and the ultimate impact on the human body.Environmental factors (including dietary habits, breastfeeding, air pollution and smoking) seem to be important in the development of IBD. However, further studies are necessary in order to fully understand these associations.Gut microbiota disbalance plays a role in IBD development. Additionally, the composition of microbiota may affect the course of the disease.There are a number of factors affecting both the risk and the course of inflammatory bowel disease. However, an increasing global prevalence of CD and UC is prompting further research aimed at gaining at better insight of these disease and improving patient care [118].

## Figures and Tables

**Figure 1 nutrients-14-02520-f001:**
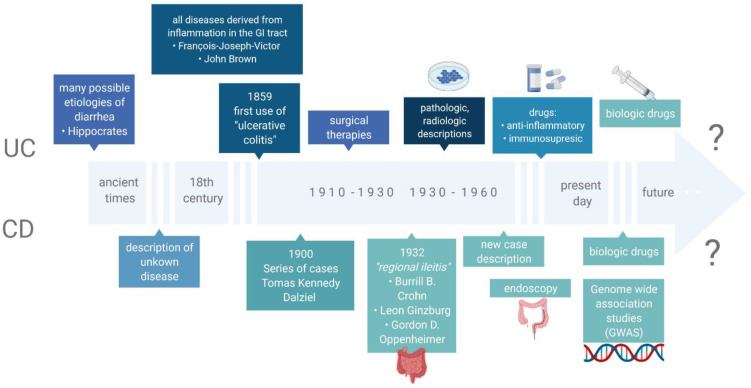
The most relevant findings related to IBD.

**Figure 2 nutrients-14-02520-f002:**
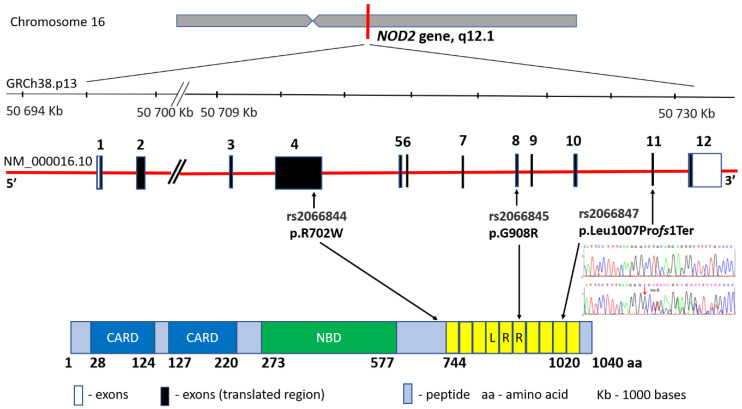
*NOD2* gene structure, main variants’ distribution and the protein product.

**Figure 3 nutrients-14-02520-f003:**
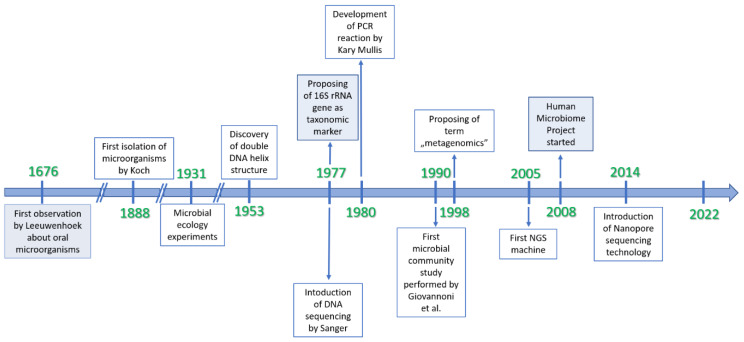
The most significant events in metagenomics over time.

**Table 1 nutrients-14-02520-t001:** IBD chromosome regions and main candidate genes.

Region Name	OMIM Number	Location	Disease	Marker (LOD Score), rs Number (*p*-Value)	Candidate Genes
IBD1	266,600	16q12	CD	D1S3669 (2.65)	*NOD2*
IBD2	601,458	12p13.2-q24.1	CD, UC	D12S83 (5.47)	*VDR*, *IFNG*
IBD3	604,519	6p21.3	CD, UC	D6S289 and D6S276 (2.07) D6S461 (4.2)	*HLA-B*, *HLA-DRB1*, *TNF*, *LTA*
IBD4	606,675	14q11-q12	CD	D14S261 (3.0)	*RNASE2*, *RNASE3*
IBD5	606,348	5q31-q33	CD	5q31-q33 (3.9)	*IL4*, *IL5*, *IL13*, *IRF1*, *CSF2*, *SLC22A4*, *SLC22A5*
IBD6	606,674	19p13	CD, UC	D19S591 (4.6)	*ICAM1*, *C3*, *TBXA2R*, *LTB4H*
IBD7	605,225	1p36	CD, UC	D1S1597 (3.01)	*TNFRSF1B*, *TNFRSF4 CASP9*
IBD8	606,668	16p	CD, UC	D16S408 (>2.5)	*IL27*, *SULT1A1*, *SULT1A2*
IBD9	608,448	3p26	CD, UC	D3S1297 (3.69)	
IBD10	611,081	2q37.1	CD		*ATG16L1*
IBD11	191,390	7q22	CD, UC	D7S669 (3.08)	*MUC3A*
IBD12	612,241	3p21.3	CD, UC	D3S2432 (1.68)	*MST1*, *BSN*, *GNAI2*
IBD13	612,244	7q21.1	CD, UC	D7S669 (3.08)	*ABCB1*
IBD14	612,245	7q32.1	CD, UC		*IRF5*
IBD15	612,255	10q21	CD, UC	rs224136 (<10 × 10^−10^)	*SIRT1*
IBD16	612,259	9q32	CD, UC	D9S2157 (1.41)	*TNFSF15*
IBD17	612,261	1p31.3	CD, UC	rs11209026 (<10^−9^)	*IL23R*
IBD18	612,262	5p13.1	CD, UC	rs1373692 (2.1 × 10^−12^)	*PTGER4*
IBD19	612,278	5q33.1	CD		*IRGM*
IBD20	612,288	10q23-q24	CD, UC	D10S547 and D10S20 (2.30)	*DLG5*
IBD21	612,354	18p11	CD, UC	rs2542151 (3.16 × 10^−8^)	*PTPN2*
IBD22	612,380	17q21.2	CD	rs744166 (6.82 × 10^−12^) rs2872507 (5.00 × 10^−9^)	*STAT3*, *ORMDL3*
IBD23	612,381	1q32.1	CD, UC	rs11584383 (1.43 × 10^−11^)	*IL10*
IBD24	612,566	20q13	CD, UC	rs2315008 (6.30 × 10^−8^) rs4809330 (6.95 × 10^−8^)	*TNFRSF6B*
IBD25	612,567	21q22.1	CD, UC	rs2836878 (6.01 × 10^−8^)	*IL10RB*
IBD26	612,639	12q15	UC	rs1558744 (2.5 × 10^−12^)	
IBD27	612,796	13q13.3	CD	rs20411 (LOD 3.98)	
IBD28	613,148	11q23.3	CD, UC		*IL10RA*

*ABCB1*: ATP binding cassette subfamily A member 1; *ATG16L1*: autophagy related 16 like 1; *BSN*: bassoon presynaptic cytomatrix protein; *CASP9*: caspase 9; CD: Crohn’s disease; *C3*: complement C3; *CSF2*: colony stimulating factor 2; *DLG5*: discs large MAGUK scaffold protein 5; *GNAI2*: G protein subunit alpha i2; *HLA-B*: major histocompatibility complex, class I, B; *HLA-DRB1*: major histocompatibility complex, class II, DR beta 1; *ICAM*: intercellular adhesion molecule, *IFNG*: interferon gamma; *IL4*, *IL5*, *IL10*, *IL13* and *IL27*: interleukins 4, 5, 10, 13 and 27, respectively; *IRF1* and *IRF5*: interferon regulatory factors 1 and 5, respectively; *IL10RA* and *IL10RB*: interleukin 10 receptor subunits alpha and beta, respectively; *IRGM*: immunity-related GTPase; LOD: logarithm (base 10) of odds; *LTA*: lymphotoxin alpha; *LTBH4H*: leukotriene B4 hydroxylase; *MST1*: macrophage stimulating 1; *MUC3*: mucin 3; *NOD2*: nucleotide binding oligomerization domain containing 2; *RNASE2*: ribonuclease A family member 2; *RNASE3*: ribonuclease A family member 3; *ORMDL3*: ORM1-like protein 3; *PTPN2*: protein tyrosine phosphatase, non-receptor type 2; *SLC22A4* and *SLC22A5*: solute carrier family 22 members 4 and 5, respectively; *STAT3*: signal transducer and activator of transcription 3; *SULT1A1* and *SULT1A2*: sulfotransferase 1A members 1 and 2, respectively; *TBXA2R*: thromboxane A2 receptor; *TNF*: tumor necrosis factor; *TNFRSF1B* and *TNFRSF4*: TNF receptor superfamily members 1B and 4, respectively; UC: ulcerative colitis; *VDR*: vitamin D receptor.

**Table 2 nutrients-14-02520-t002:** Immunological factors associated with Crohn’s disease and ulcerative colitis.

Crohn’s Disease	Ulcerative Colitis
Th1	Th2
IFNγ, IL-2, TNFα	IL-4
IL-2	IL-5
TNFα	IL-13
Il-17A	
Th17 lymphocyte and regulatory T (Treg) lymphocyte imbalance	
Il-23 polymorphism	

**Table 3 nutrients-14-02520-t003:** Environmental factors influencing the risk of inflammatory bowel disease.

Factors	Risk of Crohn’s Disease	Risk of Ulcerative Colitis
Smoking	↑	↓
Appendectomy over the age of 20	Conflicting data	↓
Oral contraceptives	↑	
Antibiotics and non-steroidal anti-inflammatory drugs (NSAIDs)	↑	
Water and air pollution	↑	
Stress	↑	
Being breastfed	↓	

↑—increase, ↓—decrease.

**Table 4 nutrients-14-02520-t004:** The most important changes in gut microbiota in inflammatory bowel disease.

Decreased Abundance	Increased Abundance
*Firmicutes*	*Ruminococcus gnavu*
*Faecalibacterium prausnitzii*	*Proteobacteria*
*Lactobacillus*	*Actinobacteria*
*Roseburia faecis*	*Escherichia coli*
*Clostridium XIVa*	*Desulfovibrio*
*Clostridium IV*	*Akkermansia muciniphila*
*Faecalibacterium prausnitzii*	*Escherichia*
*Bacteroidetes*	*Fusobacterium*
*Verrucomicrobia*	
*Anaerostipes*	
*Methanobrevibacter*	
*Faecalibacterium*	
*Peptostreptococcaceae*	
*Collinsella*	
*Christensenellaceae*	

**Table 5 nutrients-14-02520-t005:** Nutritional factors influencing the risk and/or the exacerbation of inflammatory bowel diseases.

Dietary Pattern	Risk of Crohn’s Disease	Risk of Ulcerative Colitis
High intake of saturated fatty acid	No data	↑
Fast food intake	↑	↑
Unsaturated fatty acids	Mixed results	
Sucrose	↑	↑
Vitamin C	↓	No data
Fiber	Mixed results	
Western diet pattern	↑	↑
Mediterranean diet pattern	↓	↓

↑—increase, ↓—decrease.

## Data Availability

Not applicable.

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
