# Peer review of "Where Do We Stand in the Behavioral Pathogenesis of Inflammatory Bowel Disease? The Western Dietary Pattern and Microbiota—A Narrative Review"

_nutrients, 2022, doi:10.3390/nu14122520_

Round 1

Reviewer 1 Report

The authors presented a review on environmental and nutritional genetic factors in the pathogenesis of inflammatory bowel disease.

Overall, the work addressed the individual aspects inherent to the topic of this review of scientific interest. However, the manuscript requires a great effort to make it fluent and easy to understand for the reader.

For a quick consultation of the available published data, it would be very useful for the readers to insert tables reporting the main findings described in some paragraphs.

The paragraph regarding DNA sequencing technologies and bioinformatics seems completely disconnected from the rest of the review, and in my opinion, it should be removed.

Some more recent publications on the subject are missing in this review.

Finally, the comprehension of the manuscript is extremely difficult, and a deep correction of the manuscript by a native speaker is strongly recommended.

Reviewer 2 Report

In this manuscript, Krela-kazmierczak and colleagues review the literature about the effect of the western diet in the development of IBD. The authors introduce the contribution of different factors such as genetic, environmental, and immunological factors to IBD development and summarize the effect of the western diet as well as the Mediterranean diet on IBD development. The authors highlight the research that HFD/ western diet mostly induces IBD whereas the Mediterranean diet is mostly beneficial although there are some contradictions. The manuscript is well written and timely. This will be useful for readers and researchers from a broad spectrum. However, I have the following comments.

1-( lines from 123-133), the authors should expand this section to describe the exact result. For example, the authors mentioned impaired expression of genes related to the cellular environment and junction complex….The authors should mention the exact genes and how they impact the junction complex and other parameters.

2-The microbiota shape the immune system, particularly the gut immune system. Impaired gut system in absence of particular microbiota or dysregulated immune response against microbiome induces IBD pathogenesis. The authors should include this aspect.

Line-433-472 sound redundant to the scope of this review.

3- In the conclusion section, the authors should conclude their inference more precisely and also should outline the most important aspects that need to be addressed.

4-In lines 160-168 UC is typed as CU.

Reviewer 3 Report

The review entitled “Where do we stand in the behavioral pathogenesis of inflammatory bowel disease? Western dietary pattern and microbiota - a narrative review” by Krela-Kaźmierczak et al., is a comprehensive review of the subject and will benefit the community and readers of journal nutrients.

Minor comments

1   I do have one recommendation before it is considered for publication, as the review is focused on IBD, I would recommend adding a figure with major discovery /important evets with respect to IBD disease, instead of Figure 2, which focuses on the most important events in metagenomics over time.
